# The Strength of Egg Trays under Compression: A Numerical and Experimental Study

**DOI:** 10.3390/ma13102279

**Published:** 2020-05-15

**Authors:** Leszek Czechowski, Gabriela Kmita-Fudalej, Włodzimierz Szewczyk

**Affiliations:** 1Department of Strength of Materials, Lodz University of Technology, 90-924 Lodz, Poland; 2Centre of Papermaking and Printing, Lodz University of Technology, Wólczańska 223, 90-924 Lodz, Poland; gabriela.kmita-fudalej@dokt.p.lodz.pl (G.K.-F.); wlodzimierz.szewczyk@p.lodz.pl (W.S.)

**Keywords:** finite element method, experimental studies, structure failure, egg packages

## Abstract

This work concerns the analysis of egg packages subjected to compression. Experimental investigations were carried out to determine the curves of compression and maximum loads. To compare packages accessible on the market, several different shapes of egg packages were tested after being conditioned in air with a relative humidity of 50%. Several paper structures in stock were compressed. By validating the experiment results, numerical computations based on the finite element method (FEM) were executed. The estimations of a numerical model were performed with the use of the perfect plasticity of paper and with the assumption of large strains and deflections. Our own two structures of egg packaging were taken into account: basic and modified. The material of the packages was composed of 90% recovered paper and 10% coconut fibres. This paper involved the numerical modelling of such complex packaging. Moreover, our research showed that introducing several features into the structures of the packaging can improve the stiffness and raise the maximum load. Thanks to the application of ribs and grooves, the strength ratio and compression stiffness, in comparison to the basic tray, increased by approximately 23.4% and 36%, respectively. Moreover, the obtained indexes of modified trays were higher than the majority of the studied market trays.

## 1. Introduction

Packaging is treated as a means for protecting its products from collapsing during transportation or storage; thus, it should be stiff enough for different conditions but moderately light. Owing to this fact, many packages are made of paper or paperboard to decrease their bulk and ensure their recyclability and biodegradability. An analysis of a package’s strength should be performed not only in different physical conditions but also during storing one package on another one. In other words, when they are stacked up. Apart from these elements, one important issue is the economic aspect of designing optimized structures with high mechanical properties—referred to as the weight ratio. During storage and transport, packaged eggs are in stacks. For this reason, one of the main properties of trays is their resistance to static pressure. Numerical analysis of deformations and stresses in trays subjected to loads allows predicting their load capacity but also introducing design changes that will allow achieving this load capacity with a smaller package weight, which is currently the main goal of packaging producers. Based on the literature, lately many works have involved the study of the strength of paper or paperboard packages. Fadiji et al. [1,2] analysed numerically and experimentally the strength of paperboard boxes under compression in ambient conditions and in refrigerated conditions to verify the effect of creep of packages as shown in Reference [3]. Zaheer et al. [4] also studied the strength of paperboard boxes under compression by using finite element method. The authors of Reference [5] investigated the failure of paperboard cups at deep drawing. Testing of the creasing and folding of three paperboards can be found in Reference [6]. Bai et al. [7] analysed the behaviour of the axial crushing of single corrugated paperboard. Mathematical models to predict the peak stress of several layers of corrugated paperboards were developed in Reference [8]. On the other hand, studies on composite materials in a range of buckling and post-buckling were numerically conducted in References [9,10,11,12,13] and also experimentally in References [14,15,16,17,18,19,20,21,22], among others. Analyses of progressive failures of composite structures were performed in References [23,24,25,26,27,28]. Liu et al. [29] analysed thermal-mechanical instabilities in structures by applying hybrid load-controlled and displacement-controlled algorithms. Studies on the strength of paper tubes can be found in References [30,31]. Other works referring to numerical experiments on corrugated cardboard are included in References [32,33,34,35,36]. Papers devoted to the study of honeycomb cardboards are available in References [37,38,39,40]. The authors of Reference [41] analysed the geometrically nonlinear continuum shell element for structures built of functionally graded material (FGM). They adopted, in finite elements, a simple power-law distribution function of the FGM to show the effect on the geometrically nonlinear response of the shell structures. Hernández-Pérez et al. [42] studied the twist stiffness of single- and double-wall corrugated board by using the first-order shear deformation theory. The authors of Reference [43] analysed the transverse shear properties of paper due to the short span compression test. Reference [44] investigated paperboard tubes as a formwork in bridge deck applications. This work presents the results of a series of laboratory tests conducted at the University of Wisconsin, Madison. It was stated that paperboard tube segments can be applied as formwork for certain types of slabs. Bolzon and Talassi [45] examined the behaviour of strongly anisotropic paperboard composites up to the failure by applying the burst strength testers. Borgqvist et al. [46] studied the continuum model of paperboard material by taking into account a high degree of anisotropy. Mentrasti et al. [47,48] investigated the large bending behaviour of creased paperboard experimentally and analytically. Fadiji, Coetzee, and Opera [49] studied ventilated corrugated paperboard packaging subjected to compression.

Based on the aforementioned literature, there are many studies concerning the analyses of paper, paperboards or composite materials, but there is no numerical or even experimental study on trays assigned to hold eggs. The present paper was elaborated to verify the shapes of egg packages with coconut fibres on stiffness and strength under compression. In addition, modification of egg packages was executed to improve the strength of the structures. Furthermore, the tests on other egg packages (available on the market) were carried out as well. The experimental results were related to numerical results attained by using FEM. By validation of the numerical model, a correlation between the numerical and experimental study was achieved.

## 2. Problem Description

### 2.1. Object of Analysis

The structures subjected to analysis were two different 30 egg trays with surface dimensions 300 mm × 300 mm given in Figure 1 and Figure 2. The total height of die stamping amounted to approximately 48 mm ± 0.5 mm. The first model demonstrates the basic tray (Figure 1), and the second one represents the modified tray (Figure 2). The thickness of the wall was uniform and equal to 1 mm. The main dimensions of a single cell in a tray are presented in Figure 3a,b.

### 2.2. Manufacturing Process

The die stamping built of fibres fluid by filling were used for the egg packages. The main resource for manufacturing these packages was recycled paper. Generally, the cheapest resources at lower mechanical properties were applied. The die stampings were fabricated of recycled and bio-degradable resources. The final products made of recovered paper were easy both in recycling and in utilisation. These products are simply ranked among ecological products. For the tests, 30 egg packages built of pulp in laboratory conditions were prepared. The pulp contained two grades of recovered paper easily available on the market. The first one was white office printed paper cut into strips (Figure 4a), and the second was recovered paper composed of chopped corrugated paperboards (Figure 4b).

The scrap papers were subjected to a fiberisation processing in a rotary centrifuge. To reinforce the structure of the egg packages, the coconut fibres, in an amount of 10 percent of the entire pulp, were added. Coconuts fibres are characterised by moderately high stiffness and lengths (Figure 5).

After the fiberisation processing, recovered pulp was subjected to a milling processing, the aim of which was to obtain the fibrillation, adequate ductility of fibres, and uniformity of a mass. During milling, the shortening of fibres could occur. The increase in elasticity and distribution of fibres in whole structure, thanks to the milling process, ensures an enhancement of the mechanical properties by raising the touching surfaces. The final stage of preparation was to finish the pulp by adding glue and a drainage aid. To produce the egg packaging, a transfer forming was applied. In this method, two forming dies were used. The first one was a sieve die (stable) which ensured the forming of packaging. The second one was the transfer die (movable) which enables shifting wet formed stamping to drying (Figure 6a,b). The sieve die and the transfer die were built of aluminium alloy and polypropylene, respectively (Figure 7a,b). The sieve die was covered with a steel sieve ensuring the formation of the pulp. The manufacturing process was started by supplying water to a machine vat. The water level in the machine vat amounted about to 5 cm above the sieve die to achieve equal distribution of pulp on whole sieve. Afterwards, the pulp at a concentration of 2 percent was added to the water. The consecutive stage was removing water gravitationally and sequentially under pressure of 0.05 MPa. To not damage the wet and soft tray (study samples), under pressure to the transfer die and overpressure to the sieve die were delivered. After initially drying a tray on the upper die, the formed mass was transported to a convection dryer where it reached an approximate dryness of 88–90%.

With this method, the manufactured egg trays assigned to tests differed from each other by the construction features. The egg trays were denoted as either a basic sample (Figure 8a) or a modified sample (Figure 8b).

### 2.3. Compression Test Stand

Before performing compression tests on trays, samples were dried at 40 °C and, subsequently, were conditioned according to standard PN-EN 20187:2000 (temperature 23 ± 1 °C and relative humidity 50% ± 2%).

Consequently, the samples were weighed within an accuracy of 0.001 g. The compression tests were carried out using a Zwick machine, model Z020 (Figure 9a). The range of load included from 0.1 N to 20 kN. The velocity of the compression during tests amounted to 12.5 mm/min. The arrangement of the samples in the jaws are illustrated in Figure 9b (unloaded samples) and in Figure 9c (loaded samples with initial load of 20 N).

### 2.4. FE Model

Numerical simulations were executed based on the finite element method by using MSC NASTRAN/MARC FEA 2010R^®^ version software as per Reference [50]. By applying the first-order shell finite elements (to reduce the number of degrees of freedom), discrete models were elaborated as shown in Figure 10. The size of the finite element amounted to 1 mm. The standard procedure of the density of mesh was performed by decreasing the element size until the deference in the results (at simply linear analysis without contact assumption) was below 2%. The total number of finite elements was approximately 200 k. The numerical estimations were conducted by applying Green–Lagrange equations for large strains and the second Piola–Kirchhoff stress. The number of sub-steps for the single calculations was assumed to be between 50 and 100 k. To achieve convergence, the iterations of each sub-step ranged from 10 up to 5 k.

Contact elements both between compressing plate and trays and between both trays were imposed. In the analysis, a friction coefficient value of 0.05 was taken into account. The contact detection parameter was set 0.9 (bias on tolerance). The separation criterion was based on forces. The lower plate on the external surface was fully constrained ($U_{x}=U_{y}=U_{z}=0$). The movable (upper) plate was partially constrained ($U_{x}=U_{z}=0$), where displacement of the plate was assumed $U_{y}=\mathrm{const}$. The ¼ of whole parts was modelled by imposing double symmetry conditions on symmetry planes. The boundary conditions with coordinate system are presented in Figure 10. The mechanical properties of this paper were fitted to the experimental curves based on the basic trays (simulations were performed with Young’s modulus E=100 MPa, E=120 MPa, E=150 MPa). Finally, Young’s modulus, Poisson’s ratio, yield stress, and tangential modulus were assumed as: E=100 MPa, ν=0.33, $\sigma_{0}=25\;\mathrm{MPa}$, $E^{t}=1\;\mathrm{MPa}$, respectively.

### 2.5. Statistical Analysis

During the experiment, different models of trays were studied. In all considered cases, three samples of the same build were tested (only two samples in the case of modified trays). Three samples of the basic model (EXP_1) were denoted as EXP_1_1, EXP_1_2, and EXP_1_3. The samples of modified tray (EXP_2) were denoted as EXP_2_1 and EXP_2_2. Apart from our own manufactured samples, an additional five different samples of trays taken from the market were also investigated. The samples of market trays were marked as: M_1_1, M_1_2, M_1_3 (first tray—M_1); M_2_1, M_2_2, M_2_3 (second tray—M_2); M_3_1, M_3_2, M_3_3 (third tray—M_3); M_4_1, M_4_2, M_4_3 (fourth tray—M_4); and M_5_1, M_5_2, M_5_3 (fifth tray—M_5). The composition of the market trays considered in the tests were not examined and are not known. Every studied tray of other producers differed from each other by some features and, of course, weight. Referring to the performed statistical analysis, the results of the measurements of maximum loads obtained during the compression tests of the trays were only taken into account. To eliminate the influence of sample mass on the value of the failure force, the indicators of maximum load were used. According to the method used in papermaking, the indexes were calculated as the ratio of the maximum load to the mass of the tested sample. The measurement results were subjected to a one-way statistical analysis [51] comparing the results obtained for the modified trays, EXP_2, with the results obtained for the other trays. The distribution of Student’s t-test statistics with a significance level of α = 0.05 was adopted.

## 3. Results and Discussion

### 3.1. Results of the Study

This subsection shows both the experimental and numerical results for the trays. The tests were conducted for egg packages after being conditioned in air with a relative humidity of 50%. Figure 11 presents the compression force versus shortening for basic trays (denotation: EXP_1_1, EXP_1_2, and EXP_1_3). The numerical study was performed for several Young’s moduli in the case of the basic model (FEM_1) to fit to the empirical curves. It turned out that the Young’s modulus corresponding to 100 MPa was the closest to the experimental characteristics, and for further calculations this value was assumed. In the case of curves obtained by FEM, some fluctuations were observed. It was probably caused due to the sudden lateral shifts of some cells in the first tray with reference to the other cells in the second tray (temporarily slight drops or growths in stiffness of numerical models (Table 2), where some deformations of trays are sequentially presented). This phenomenon was not visibly registered in experiment. The next chart (Figure 12) illustrates the compression forces for modified trays (denotation: EXP_2_1 and EXP_2_2). The experiment for modified trays was conducted only for two specimens. In the case of basic trays, the average mass amounted to 54.3 g. The mean value of the strength ratio (defined as maximum force referred to mass of tray) was 31.5 N/g. The stiffness recorded during compression within 30% of total load ranged from 247.5 N/mm to 363.3 N/mm (average value as 314.7 N/mm). Taking into account modified trays, the average strength ratio was equal to 38.9 N/g. Furthermore, average compression stiffness also increased (for EXP_2_1 and for EXP_2_2, 396.5 N/mm and 459.8 N/mm, respectively).

Based on the results, the mean stiffness of the modified structure increased by 36% and the maximum mean force amounted to approximately 2418 N in the case of the experiment (increased by 41.0% with respect to the basic tray). In the case of the strength ratio, the average rise in comparison to the basic tray was noticed by 23.4%. The increase in the parameters can be caused apparently due to the applied changes in the structure of the trays as additional ribs and grooves (See Figure 2 and Figure 3b). In the case of the numerical results, the maximum force was slightly lower and equal 1922 N (FEM_2). To check the stiffness and strength of the trays accessible on the market, five other trays from different producers were analysed. The names of producers are not given with respect to the authors’ rights. The compression tests on the trays from the market were carried out a minimum of 3 times (results are given for three samples). The curves compared to the tested trays (curves for chosen samples) are plotted in Figure 13. Denotation of these trays are given in Section 2.5. All characteristics’ values are shown in Table 1, where Δ represents the standard deviation (for mass $\Delta_{m}$, for stiffness $\Delta_{S}$, for maximum load $\Delta_{F}$, and for strength ratio $\Delta_{\mathrm{SR}}$).

Two cases in which the test probability was lower than the assumed significance level p<0.05 are marked in Table 1 by the letters ${}^{a}$ and ${}^{b}$ in superscript for the trays EXP_1^a^ and M_2^b^ compared with the modified trays.

The conclusions resulting from the statistical analysis confirm that the mean maximum load obtained for the modified tray EXP_2 was significantly different from the mean maximum load attained for the basic tray EXP_1 and market tray M_2 at the significance level of 95%. In the case of comparing the mean maximum load carried by the modified trays and trays available on the market, M_1, M_ 3, M_ 4, M_5, the statistical test showed the possibility of insignificant differences among the compared mean values of measurements. In such cases, it can be assumed that the modified trays were not weaker than market trays from which the basic trays were weaker. Based on the results for the market trays, comparable values in a comparison to our own manufactured trays were obtained. Namely, the average compression stiffness was between 283.4 N/mm (for samples M_2) and 354.3 N/mm (for samples M_4). For these trays, the mean maximum load ranged from 1951 N (M_1) to 2353 N (M_4). Moreover, also for M_4, the average strength ratio turned out to be the greatest among all considered cases (43.6 N/g). In comparison to the modified trays, the mean strength ratio of the M_4 samples was higher by 12.1%, but the mean maximum loads was the greatest for modified trays. Registered parameters for modified trays were mostly also better than measured parameters of other market trays (See Table 1). In the case of M_3 and M_5, the greatest standard deviations, $\Delta_{F}$, of maximum loads were observed ($\Delta_{F}=\pm 283\;N$ and $\Delta_{F}=\pm 422\;N$, respectively). Taking into considered our own produced trays, standard deviations of maximum loads, $\Delta_{F}$, were meaningfully smaller than the market trays and equal to $\Delta_{F}=\pm 100\;N$ and $\Delta_{F}=\pm 11\;N$ for basic trays and modified trays, respectively. In the case of compression stiffness, the standard deviations, $\Delta_{S}$, were comparable for all considered trays (up to a few dozens). The greatest standard deviation of the strength ratio, $\Delta_{\mathrm{SR}}$, was observed for the samples for M_5 ($\Delta_{\mathrm{SR}}=\pm 8.5\;N/g$).

### 3.2. Maps of Stresses

The distribution of the effective stresses on both deformed models are presented in Table 2 (for E=100 MPa). In the case of FEM_1, the maximum stresses on any point in the structure during compression amounted to 6.01 MPa (point 6), 15.9 MPa (point 7), 18.7 MPa (point 8), 20.5 MPa (point 9), and 20.5 MPa (point 10).

Taking into account modified trays (FEM_2), maximum effective stresses were obtained: 13.3 MPa (point 1), 17.6 MPa (point 2), 18.1 MPa (point 3), 18.3 MPa (point 4) and 18.9 MPa (point 5). Based on deformations maps, it can be seen that behaviours of each structure differed from each other. It was noticed that in comparison to basic trays (FEM_1), the modified structures (FEM_2) carried the load rather uniformly. It means that structure deforms largely in many regions (points 3–5) which apparently can be profitable to increasing the stiffness and strength of the trays.

## 4. Summary

This paper concerns the strength analysis of egg trays subjected to compression. Compression tests were conducted for a pair of stacked trays. To compare the results, the experiments were performed for other trays accessible on the market. Based on the results, it was confirmed that the modified package structure, thanks to the application of ribs and grooves, were stiffer and stronger than the basic model or the other examined trays, although their composition was not known.

Comparing the values of the maximum forces obtained during the measurement of basic trays and modified trays, we observed an increase in force by approximately 41% of the value obtained for a basic tray. An upward trend can also be observed in the case of the strength ratio (strength-to-weight ratio) by approximately 23% for a modified tray, which indicates the impact of design changes on the load-carrying capacity of the trays.

## Figures and Tables

**Figure 1 materials-13-02279-f001:**
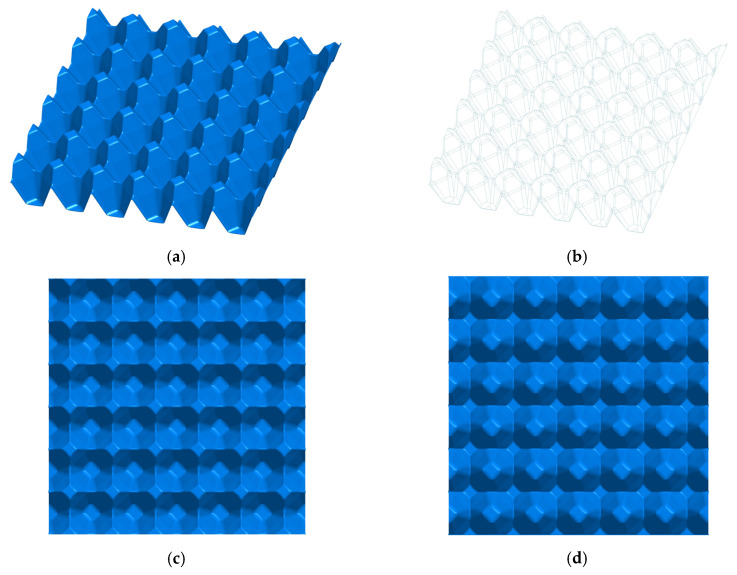
A 3-dimensional model of a basic tray (**a**,**b**), view from the top (**c**), and view from the bottom (**d**).

**Figure 2 materials-13-02279-f002:**
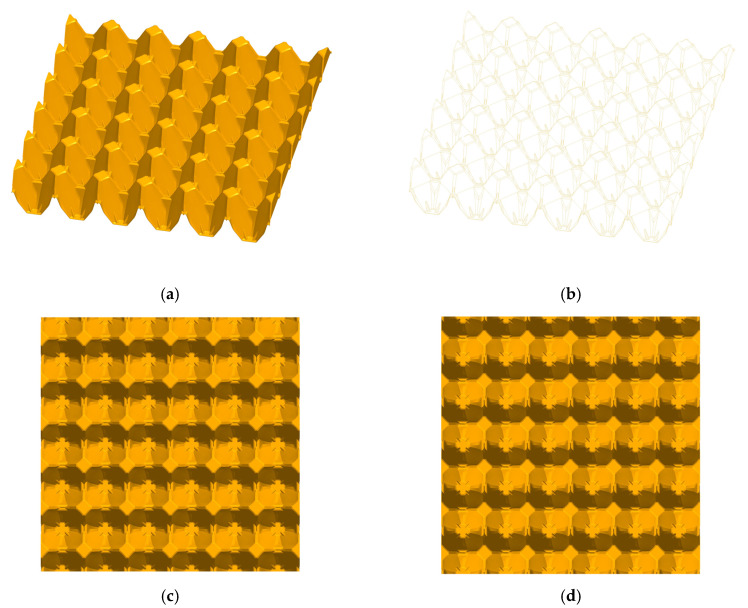
A 3-dimensional model of modified tray (**a**,**b**), view from the top (**c**), and view from the bottom (**d**).

**Figure 3 materials-13-02279-f003:**
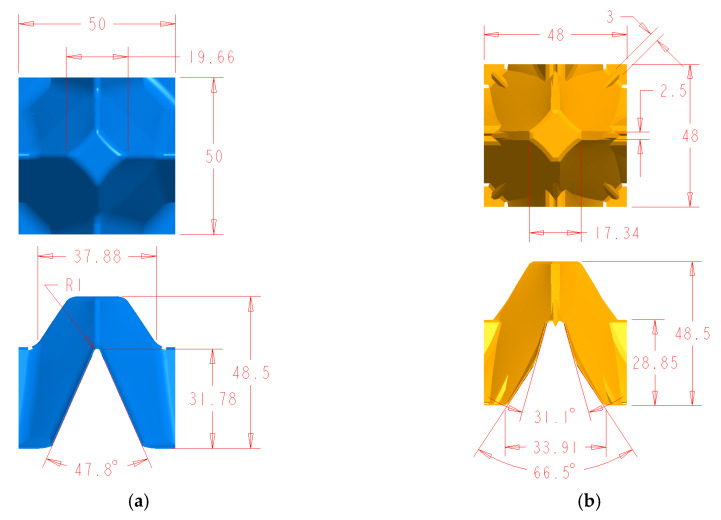
The drawing of the basic model (**a**) and of the modified model (**b**).

**Figure 4 materials-13-02279-f004:**
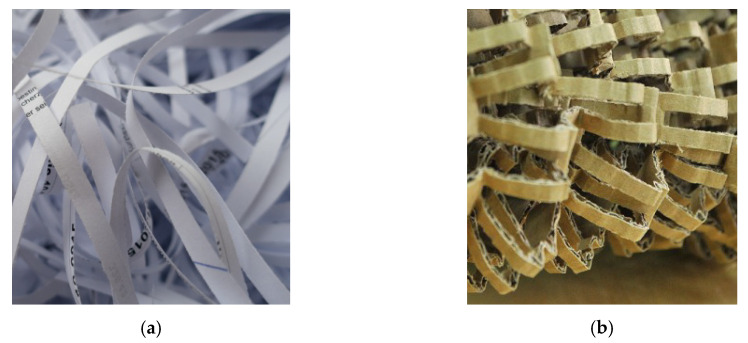
The grades of recovered paper: office wastepaper (**a**), chopped corrugated paperboards (**b**).

**Figure 5 materials-13-02279-f005:**
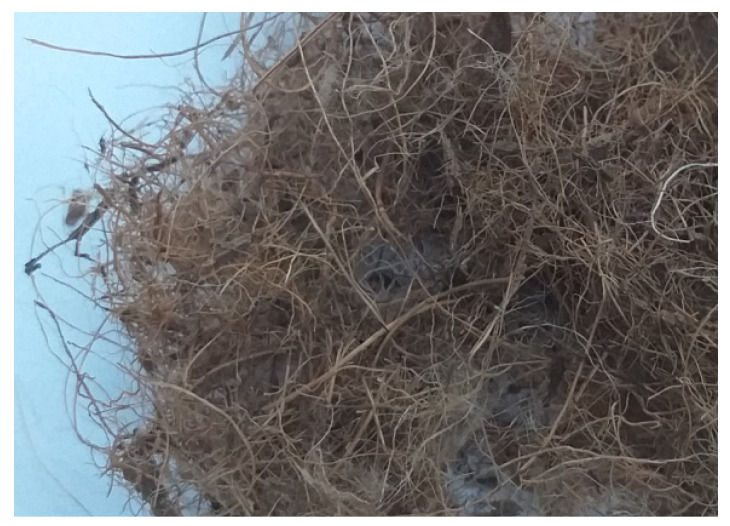
The fibres of coconuts.

**Figure 6 materials-13-02279-f006:**
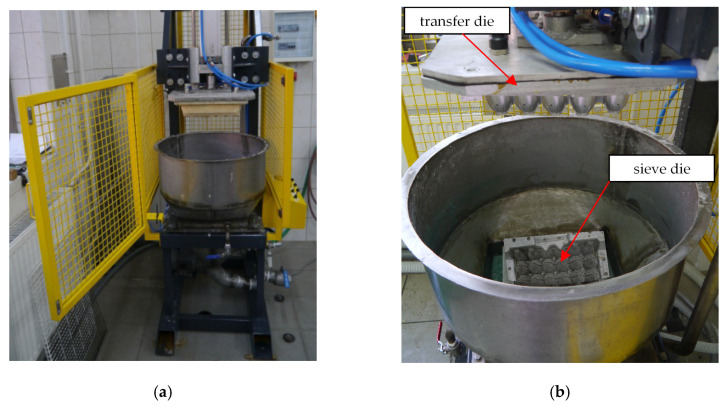
View of the entire forming machine (**a**) and a view of the forming dies (**b**).

**Figure 7 materials-13-02279-f007:**
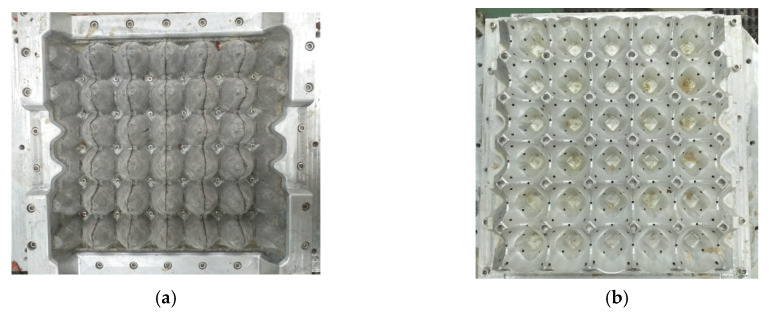
View of the sieve die (**a**) and view of the transfer die (**b**).

**Figure 8 materials-13-02279-f008:**
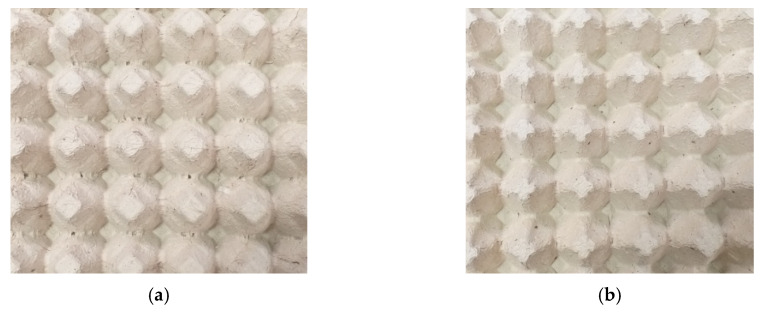
View of a basic tray (**a**) and a modified tray (**b**) assigned to tests.

**Figure 9 materials-13-02279-f009:**
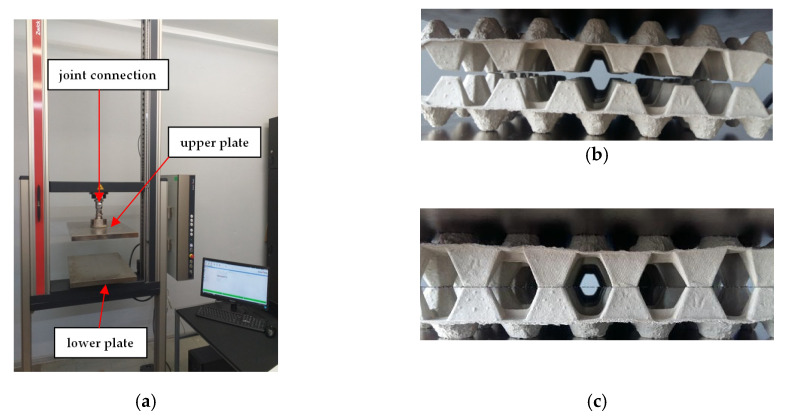
Zwick machine (**a**), unloaded trays in jaws before the test (**b**), and arrangement of trays under initial load (**c**).

**Figure 10 materials-13-02279-f010:**
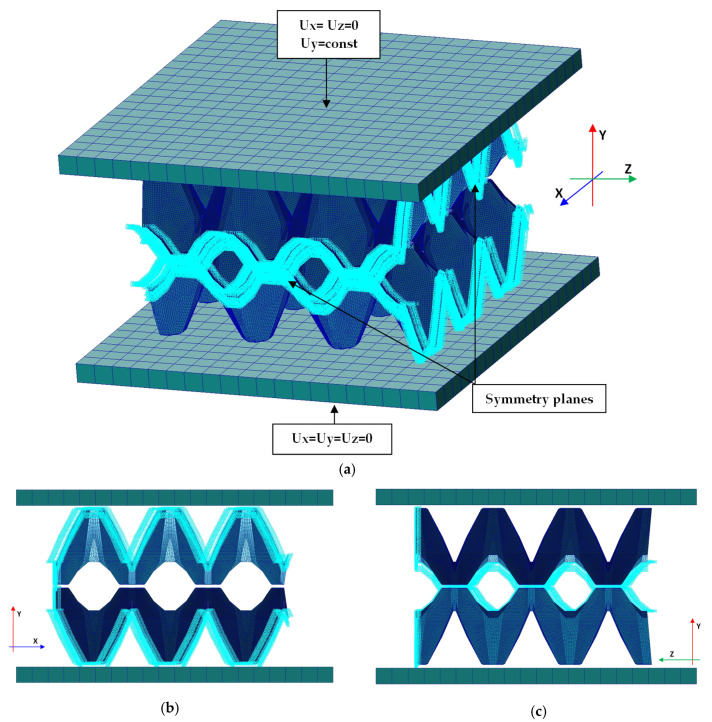
Numerical models with boundary conditions (**a**), view of discrete model in the plane X —Y (**b**) and view of discrete model in the plane Y—Z (**c**).

**Figure 11 materials-13-02279-f011:**
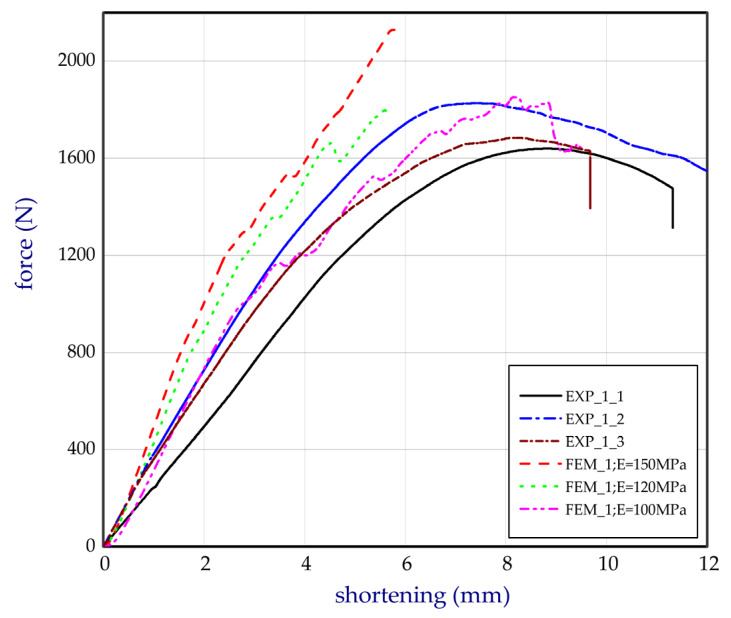
Total force of compression in a function of shortening for the basic model.

**Figure 12 materials-13-02279-f012:**
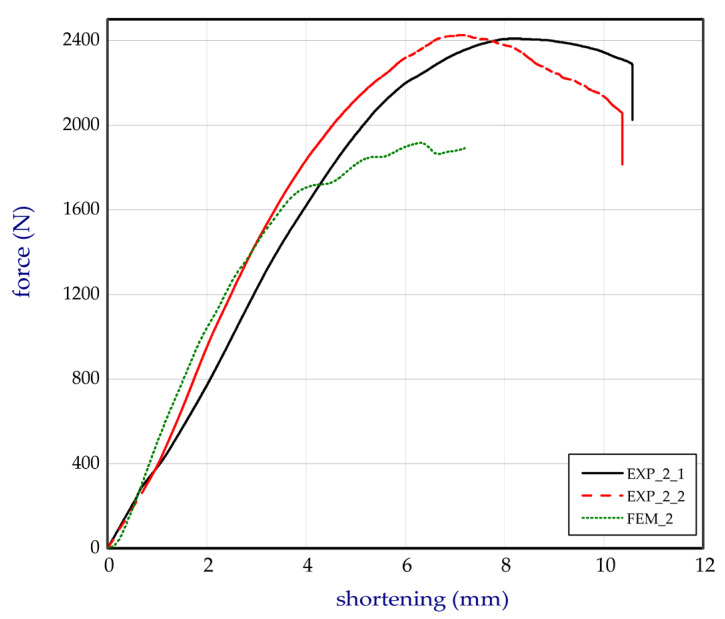
Total force of compression in a function of shortening for the modified model.

**Figure 13 materials-13-02279-f013:**
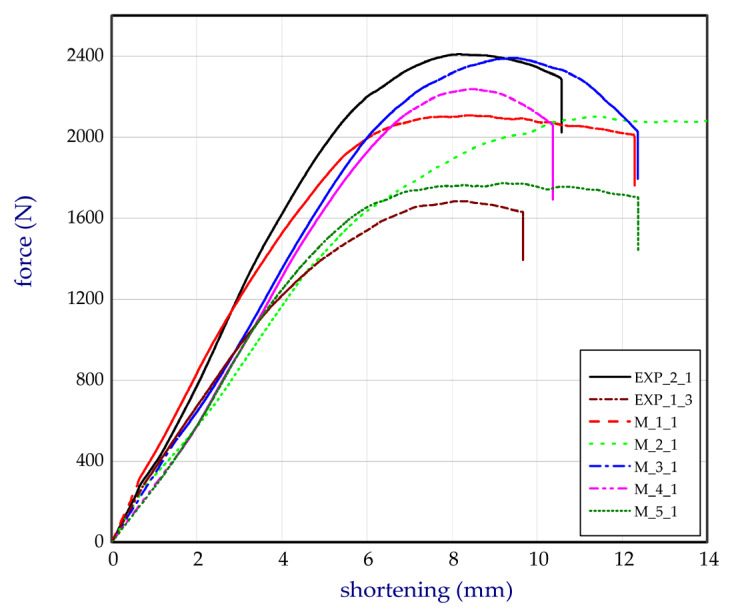
Comparison of curves (force versus shortening) based on different models.

**Table 1 materials-13-02279-t001:** Results of characteristic parameters (* average values for three specimens, ** average values for two specimens).

Denotation of Sample/ Model	Mass of Tray (Mean Mass of Tray ± $\Delta_{m}$ (g)	Stiffness of Tray (Mean Stiffness of Tray ± $\Delta_{S}$ (N/mm)	Maximum Load (Mean Maximum Load ± $\Delta_{F}$ (N)	Strength Ratio (Mean Strength Ratio ± $\Delta_{SR}$ (N/g)	Decrease (−)/Increase (+) of Mean Stiffness with Respect to Basic Tray (Modified Tray) (%)	Decrease (−)/Increase (+) of Mean Maximum Load with Respect to Basic Tray (Modified Tray) (%)	Decrease (−)/Increase (+) of Strength Ratio with Respect to Basic Tray (Modified Tray) (%)
EXP_1_1 ^a^	52 (54.3 * ± 2.1)	247.5 (314.7 * ± 60.1)	1634 (1715 * ± 100)	31.4 (31.5 * ± 1.0)	0 (−26.5)	0 (−29.1)	0 (−19.0)
EXP_1_2 ^a^	56 (54.3 * ± 2.1)	363.3 (314.7 * ± 60.1)	1827 (1715 * ± 100)	32.6 (31.5 * ± 1.0)	0 (−26.5)	0 (−29.1)	0 (−19.0)
EXP_1_3 ^a^	55 (54.3 * ± 2.1)	333.2 (314.7 * ± 60.1)	1685 (1715 * ± 100)	30.6 (31.5 * ± 1.0)	0 (−26.5)	0 (−29.1)	0 (−19.0)
FEM_1	-	361.1	1830	-	-	-	-
EXP_2_1	60 (62 ** ± 2.8)	396.5 (428.2 ** ± 44.8)	2410 (2418 ** ± 11)	40.2 (38.9 ** ± 1.6)	+36.0 (0)	+41.0 (0)	+23.4 (0)
EXP_2_2	64 (62 ** ± 2.8)	459.8 (428.2 ** ± 44.8)	2426 (2418 ** ± 11)	37.9 (38.9 ** ± 1.6)	+36.0 (0)	+41.0 (0)	+23.4 (0)
FEM_2	-	496.2	1922	-	-	-	-
M_1_1	60 (61 * ± 3.1)	296.3 (335.8 * ± 35.6)	2107 (1951 * ± 136)	35.1 (32.2 * ± 2.8)	+6.71 (−21.6)	+13.8 (−19.3)	+2.22 (−17.2)
M_2_1 ^b^	70 (71 * ± 2.6)	284.4 (283.4 * ± 15.8)	2087 (2090 * ± 77)	29.8 (29.4 * ± 0.3)	-9.99 (−33.8)	+21.9 (-13.7)	-6.66 (−24.2)
M_3_1	62 (60 * ± 2.1)	321.3 (341.8 * ± 22.9)	2391 (2067 * ± 283)	38.6 (34.2 * ± 3.8)	+8.61 (−20.2)	+20.5 (−14.5)	+8.57 (−12.0)
M_4_1	55 (54 * ± 2.6)	307.7 (354.3 * ± 51.8)	2237 (2353 * ± 141)	40.7 (43.6 * ± 2.6)	+12.6 (−17.3)	+27.1 (−2.69)	+38.4 (+12.1)
M_5_1	61 (58 * ± 2.5)	289.9 (324.1 * ± 47.8)	1766 (2251 * ± 422)	29.0 (38.8 * ± 8.5)	+2.98 (−24.3)	+31.3 (−6.90)	+23.2 (−0.26)

**Table 2 materials-13-02279-t002:** Von Mises stress maps for FEM_1 (on the bottom) and FEM_2 (on the top).

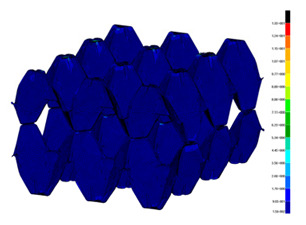	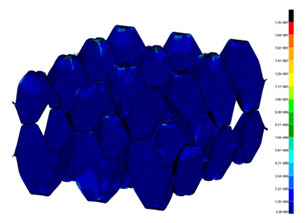	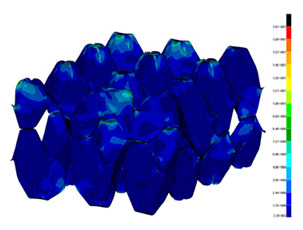	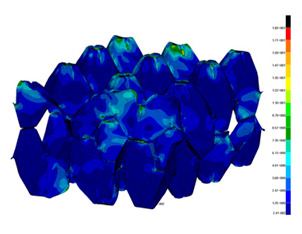	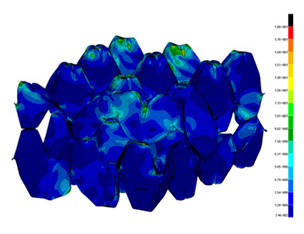
point 1	point 2	point 3	point 4	point 5
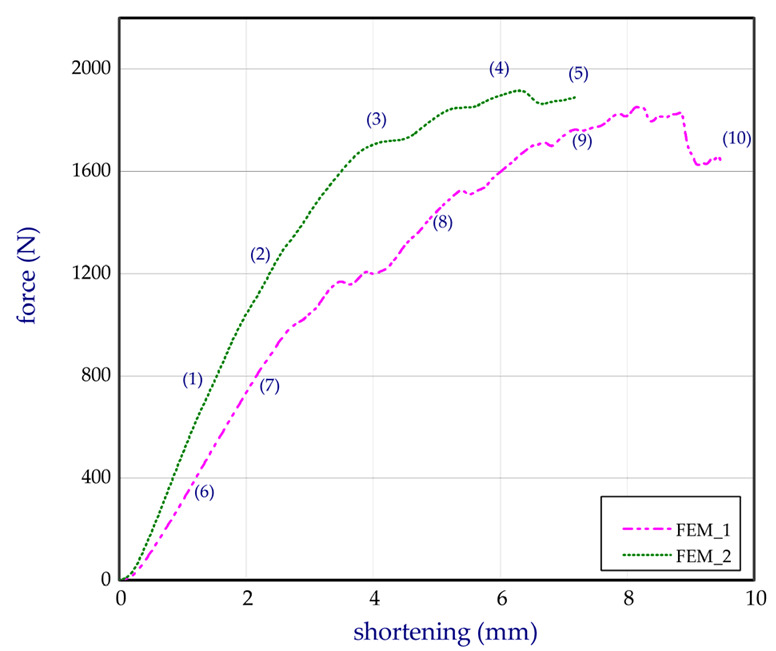				
point 6	point 7	point 8	point 9	point 10
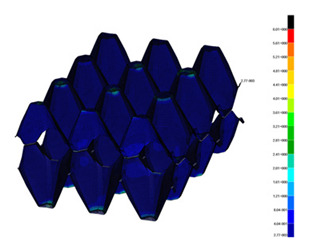	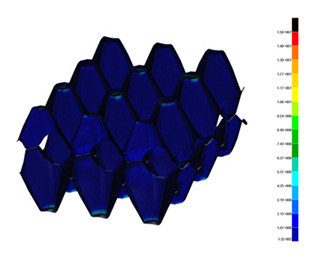	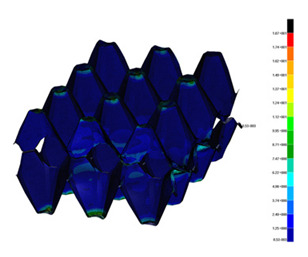	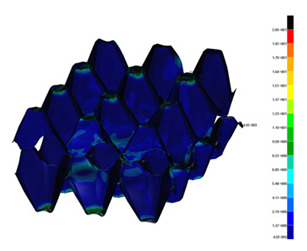	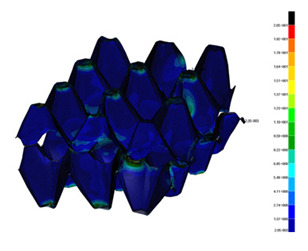

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
