# Peer review of "The Strength of Egg Trays under Compression: A Numerical and Experimental Study"

_materials, 2020, doi:10.3390/ma13102279_

Round 1
Reviewer 1 Report
The manuscript deals with the strength of the eggs trays under compression – numerical and experimental study.
Please separate values from units, e.g. “100 MPa” not “100MPa”.
Abstract
This section is vague. Please add your main results.
Materials and methods
Testing also each package with eggs??? The eggs will also have some effect on the behavior of each package???
%moisture of each package???
A statistical analysis section is missing.
Results and discussion
Line 165- “All characteristics values are shown in Table 1. In case of basic trays, the average mass amounted to 54.3 g. The mean value of the strength ratio (defined as maximum force referred to mass of tray) was 31.6 N/g. The stiffness recorded during compression within 30% of total load ranged from 247.5 N/mm to 363.3 N/mm (average value as 314.6 N/mm).”???Table 1, Please add average values plus standard deviation. Moreover, please add different superscript letters for significant differences and revise the discussion in accordance.
Summary
Line 200- “Based on results, it was confirmed that modified package structure thanks to applications some ribs and grooves can be stiffer and stronger than basic model or other examined trays though their composition wasn’t known. Comparing the values of the maximum forces obtained during the measurement of basic trays and modified trays, we observe an increase in force by approx. 40% of the value obtained for basic tray. An upward trend can also be observed in the case of the strength ratio (strength to weight ratio) by approx. 20% for a modified tray, which indicates the impact of design changes on the load capacity of the trays.”???what is the main goal regarding the tested package???egg protection. This part must be clarified.
Author Response
Dear Reviewer,
first of all, we’d like to thank for review.
The responses are included in attached file.
Best Regards,
Leszek Czechowski
Gabriela Kmita-Fudalej
WÅ‚odzimierz Szewczyk

Reviewer 2 Report
The manuscript discusses the experimental tests and numerical modeling of egg trays subject to compression load. Elastic perfectly plastic material assumption is used and large deformation is considered. The authors also suggested improvements on the structural stiffness. The manuscript is well organized, and the topic falls within the scope of the journal. Before recommending publication, I would like to ask the authors to address the remarks listed below.
(1) In the numerical model, how did the authors come up with the finite element size of 1 mm? Did the authors do a mesh sensitivity study and a convergence analysis?
(2) The use of the Green strain only allows for medium to large strain. It does not necessarily guarantee large displacement. Did the authors consider geometric nonlinearity in the FE analysis? This is very important because the failure pattern is dominated by buckling.
(3) More details should be provided to ensure the reproducibility. For example, the authors mentioned that contact elements were imposed between the trays and the plates. What type of contact did the authors use (e.g., surface to surface, node to surface)? Is there contact definition between the two trays as well?
(4) What is the justification for using first-order shell finite elements? In other words, why didn’t the authors consider using higher-order shell elements such as the one discussed here (doi.org/10.1016/j.compstruct.2020.111893)? A short discussion on the linear shell decision can be included.
(5) In the comparison between the FE model and the experimental tests, what is the cause of multiple sudden drops in stiffness, especially in the purple curve? The authors need to include the explanation in the manuscript.
(6) As the authors realized, sudden buckling/instability is the mean cause of the package failure, and therefore it is important to include some buckling/instability literature in the introduction section. Papers that can be considered are as follows: (1) doi.org/10.1061/(ASCE)EM.1943-7889.0001263
(7) There exist a large number of grammar errors throughout the manuscript, such as “eggs package” (should be “egg package”, “egg trays”), “as a mean for” (should be “as a means for”), and “other words” (should be “in other words”). The writing should be further polished as well. For example, the reviewer does not understand what the authors mean by “The paper reveals the approach of numerical modeling such a complex packages”.
Author Response

(The authors gave the same response as above.)

Round 2
Reviewer 1 Report
Materials and methods
A statistical analysis section is missing.
Results and discussion
Lines 216-220- Please separate values from units, e.g. “100 MPa” not “100MPa”.
Table 1- Please add different superscript letters for significant differences and revise the discussion in accordance.
Author Response
Dear Reviewer,
once again thank you for review of our paper.
The added/changed text has been in yellow highlighted.
With best Regards,
Leszek Czechowski
Gabriela Kmita-Fudalej
WÅ‚odzimierz Szewczyk

Reviewer 2 Report
I don't have any further comment to the manuscript.
Author Response

(The authors gave the same response as above.)

Round 3
Reviewer 1 Report
Materials and methods
In the statistical analysis section, the authors must present the statistical tests that were performed to the experimental results. i.e. ANOVA used??then, Post-hoc test used??Tukey’s??LSD???significance level used??
Results and discussion
Table 1- In this table the authors must identify by different superscript letters (a, b etc…) all the pair(s) of results that are significantly different (p<0.05), and revise the discussion in accordance.
Author Response
Dear Reviewer,
the text has been corrected based on suggestions of Reviewer. In case of acceptance before publishing, the English in manuscript will be checked finally by Proofreader.
Best Regards,
Leszek Czechowski
Gabriela Kmita-Fudalej
WÅ‚odzimierz Szewczyk
